# Protease-associated import systems are widespread in Gram-negative bacteria

Rhys Grinter[1,2,3]*, Pok Man Leung[1], Lakshmi C. Wijeyewickrema[4], Dene Littler[2], Simone Beckham[2,5], Robert N. Pike[4], Daniel Walker[6], Chris Greening[1], Trevor Lithgow[2]*

1 School of Biological Sciences, Monash University, Clayton, Victoria, Australia, 2 Infection and Immunity Program, Biomedicine Discovery Institute and Department of Microbiology, Monash University, Clayton, Australia, 3 Institute of Microbiology and Infection, School of Immunity and Infection, University of Birmingham, Birmingham, England, United Kingdom, 4 Department of Biochemistry and Genetics, La Trobe Institute of Molecular Sciences, La Trobe University, Melbourne, Victoria, Australia, 5 La Trobe Rural Health School, College of Science, Health and Engineering, La Trobe University, Bendigo, Australia, 6 Institute of Infection, Immunity, and Inflammation, College of Medical, Veterinary, and Life Sciences, University of Glasgow, Glasgow, United Kingdom

* rhys.grinter@monash.edu (RG); trevor.lithgow@monash.edu (TL)

## Abstract

Bacteria have evolved sophisticated uptake machineries in order to obtain the nutrients required for growth. Gram-negative plant pathogens of the genus *Pectobacterium* obtain iron from the protein ferredoxin, which is produced by their plant hosts. This iron-piracy is mediated by the ferredoxin uptake system (Fus), a gene cluster encoding proteins that transport ferredoxin into the bacterial cell and process it proteolytically. In this work we show that gene clusters related to the Fus are widespread in bacterial species. Through structural and biochemical characterisation of the distantly related Fus homologues YddB and PqqL from *Escherichia coli*, we show that these proteins are analogous to components of the Fus from *Pectobacterium*. The membrane protein YddB shares common structural features with the outer membrane ferredoxin transporter FusA, including a large extracellular substrate binding site. PqqL is an active protease with an analogous periplasmic localisation and iron-dependent expression to the ferredoxin processing protease FusC. Structural analysis demonstrates that PqqL and FusC share specific features that distinguish them from other members of the M16 protease family. Taken together, these data provide evidence that protease associated import systems analogous to the Fus are widespread in Gram-negative bacteria.

## Author summary

To grow and cause infection bacteria must obtain essential nutrients from their environment or host. The element iron is one such nutrient and is often contained inside proteins, the building blocks of hosts cells. Bacteria that cause disease in plants are able to extract iron from plant proteins, by importing the protein and cutting it up once inside the bacterial cell. While it was known that specific bacteria that infect plants can do this, it was unclear if other bacteria that infect humans and animals are also able to import host proteins. In this work we analysed the genetic sequences of bacteria and found that genes

**Data Availability Statement:** The crystallographic coordinates and associated structure factors produced in this study are available in the Protein Data Bank (PDB) with the accession codes YddB =

6OFR, PqqL Full Length = 6OFS, PqqL N-terminal Domain = 6OFT. Small angle X-ray scattering data for PqqL is available in the SASBDB accession code SASDFB6.

**Funding:** The work was funded by the Australian Research Council (ARC; FL130100038) https://www.arc.gov.au/. R.G. was funded by a Sir Henry Wellcome Fellowship award (106077/Z/14/Z) https://wellcome.ac.uk/. T.L. is an ARC Australian Laureate Fellow (FL130100038). The funders had no role in study design, data collection and analysis, decision to publish, or preparation of the manuscript.

**Competing interests:** The authors have declared that no competing interests exist.

responsible for importing and processing proteins are widespread in bacteria that cause disease in humans, animals and plants. We analysed the structure and chemistry of the protein products of these genes and found that they possess characteristics that are necessary and sufficient for importing and processing proteins. Our conclusion from this work is that the ability to import host proteins to gain nutrients is common in bacteria.

## Introduction

Bacteria often experience a scarcity of the resources they require to grow, divide and persist [1]. In many environments this is due to competition with other microorganisms, while during infection of plants or animals, the host employs strategies to deny bacteria essential nutrients to prevent their growth [2]. Iron-limitation is a key host defence strategy and, in order to overcome this, infectious bacteria have evolved sophisticated iron uptake machinery [3].

It was recently shown that the Gram-negative phytopathogens *Pectobacterium carotovorum* and *Pectobacterium atrosepticum* are able to obtain the essential nutrient iron from the plant-protein ferredoxin [4, 5]. In *Pectobacterium*, iron acquisition *via* plant ferredoxin is mediated by the Ferredoxin uptake system (Fus), a molecular machine consisting of inner and outer membrane transporters and a periplasmically localised protease [6–9]. Intriguingly, the outer membrane transporter from this system, a TonB-dependent transporter (TBDT) designated FusA, imports intact ferredoxin into the periplasm of the bacteria where it is processed by the M16 family protease FusC [6–8]. This is the first example of a bacterium importing an intact protein for nutrient acquisition, with previously described extraction of protein cofactors taking place on the bacterial cell surface [10, 11]. It is also remarkable considering the transported ferredoxin has dimensions barely smaller than the internal pore of FusA [7]. Proteolytic cleavage of ferredoxin by FusC in the periplasm, results in the release of its iron-sulphur cluster [8], which it is hypothesized is transported into the bacterial cytoplasm by the inner membrane transporter FusD [6].

The observation that bacteria import and process ferredoxin for nutrient acquisition is unprecedented [10, 11]. It was unclear, however, whether this ability is specific to *Pectobacterium* or a more common strategy implemented by Gram-negative bacteria. To address this question, we interrogated available bacterial genomes for sequences related to the outer membrane transporter FusA. This search showed that gene clusters resembling the Fus are widespread across Proteobacteria and are present in bacteria that adopt a variety of different lifestyles, including many bacteria that form an association with plant or animal hosts. The composition of these gene clusters supports a broad role in protein import, with FusA genes commonly associated with putative M16 processing proteases. To confirm the common architecture of these systems, we characterised the gene cluster analogous to the Fus from *Escherichia coli*: the ydd/pqqL operon. This showed that despite their distant relationship to the Fus from *Pectobacterium*, Ydd/PqqL shares a common structure, localisation and regulation. In combination with previous studies, these data provide evidence that protein import systems related to the Fus represent a novel family that are widespread in Proteobacteria, where they may function in obtaining iron from host proteins.

## Results

### Gene clusters related to the Fus are widespread in Proteobacteria

The sequence of FusA was used to interrogate the UniProtKB database using the HMMER search algorithm [12]. Through this approach, we were able to select the sequences of various

FusA-related proteins while excluding other members of the wider TBDT protein superfamily (S1 Fig & S1 Table). Proteins analogous to FusA were identified in a wide variety of species throughout the Proteobacteria. These sequences were clustered via pairwise similarity scores using the program CLANS, forming 24 sequence clusters (Fig 1) [13]. While present in a wide variety of bacteria, FusA homologues were not universally distributed across bacterial clades. For example, *Escherichia coli* possesses YddB, a distant homologue of FusA (24% amino acid identity), whereas FusA homologues were not detected in closely related genera like *Salmonella* and *Citrobacter*. Another striking observation was the diversity of FusA homologues, with the amino acid identity of sequences within CLANS clusters ranging from ~ 30 to 100%. As proteins within each cluster are more closely related to each other than other FusA homologues, the overall diversity of the FusA protein family is very high (S1 Table).

Due to its confirmed role in protein processing upon import [6, 8], the presence of FusC genetically associated with the FusA homologues was examined (Figs 1 and 2). FusC homologues associated with FusA were identified in 12 of the 24 sequence clusters, including the four largest clusters, which we defined by members of the genera predominantly found within the cluster: *Pectobacterium*, *Pseudomonas*, *Haemophilus* and *Providencia* (Fig 1, S1 Table). An association between FusA and FusC homologues is not restricted to closely related clusters, nor is the presence of a FusC homologue universal to all members of a cluster (Figs 1 and 2 and S1 Table); the stochastic distribution of FusC homologues suggests a related function, rather than evolutionary conservation alone, drives their genetic association with FusA homologues.

The composition of Fus gene clusters inside and between sequence groups was diverse, consisting of different arrangements of FusA and FusC homologues and a variety of other genes with a possible function in iron acquisition and protein import (Fig 2). In some cases, genes encoding other proteases were associated with FusA homologues, suggesting different processing factors may be employed by some systems. In multiple cases where FusC homologues were absent, genes encoding members of the cytochrome *c* peroxidase family were identified. These dual-haem containing enzymes are redox active [14, 15] and thus may be responsible for reducing an imported substrate protein, inducing it to release an iron containing cofactor without proteolytic processing.

Genome metadata was mined to determine the environment from which the bacteria had been isolated, showing they adopt a variety of lifestyles, which tend to correlate with sequence cluster (S1 Table). For example, members of the clusters defined by *Pectobacterium* and *Hemophillus* sequences adopt a commensal or pathogenic relationships with plant or animal hosts, while members of the *Marinomonas*, *Marinobacter* and *Pseudoalteromonas* clusters were isolated from marine or other environmental samples (S1 Table). This suggests that different lineages of these Fus homologues provide an advantage to bacteria adopting specific lifestyles.

Taken together, these data show that homologues of FusA are widespread in Proteobacteria, but their distribution is sporadic, consistent with them providing niche- or lifestyle-specific functions. The presence of associated FusC homologues, or other proteases, in a large number of widely distributed sequence clusters suggests a general role for these Fus gene clusters in protein import and processing.

## YddB is structurally analogous to FusA and possesses a large external substrate binding site

In order to determine if FusA homologues possess the structural features required for protein import, we purified and characterised YddB, the FusA homologue from *E. coli* (Fig 3A). YddB belongs to the outer membrane localised TBDT family and has been detected in the outer

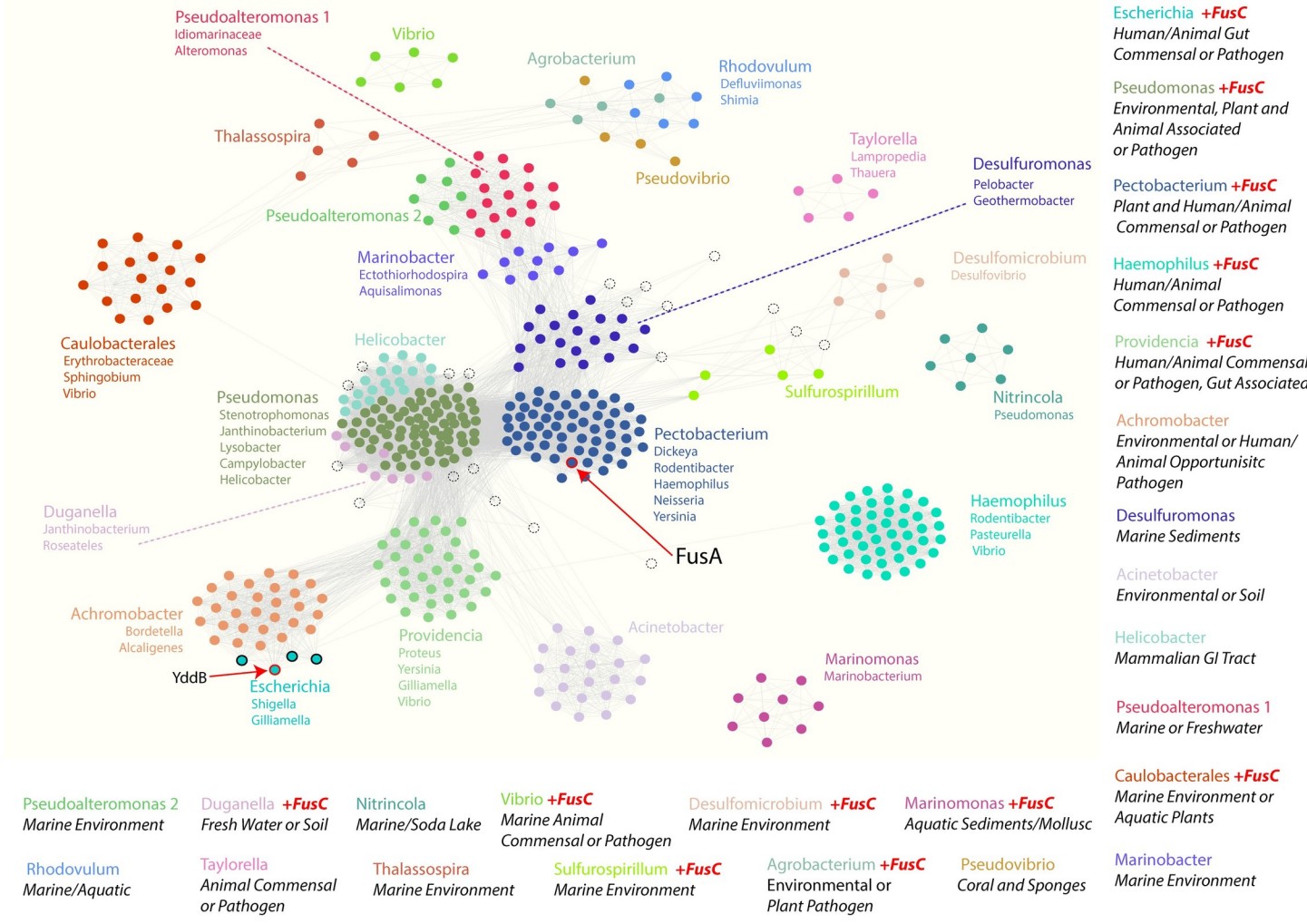

**Fig 1. Clustering analysis of FusA homologues.** CLANS similarity network analysis identifies clusters of sequences in the FusA homologues identified by HMMER search of the UniProt reference proteomes dataset. Clusters are named based on the genus of a major species of origin for the cluster, with members of other prevalent genera listed. The presence of FusA sequences associated with the homologues of the protease FusC inside a cluster are noted, as is the most common lifestyle for bacteria in the cluster. Dots represent individual sequences and grey lines represent pairwise similarity relationships. An E-value cut-off of 1e-110 was used for clustering. A full list of sequences and metadata where available is provided in S1 Table.

membrane and in outer membrane vesicles of *E. coli* in a number of studies [16–20]. YddB is encoded in an operon with the FusC homologue PqqL, which has been shown to be expressed in response to iron limitation, to be regulated by the ferric uptake regulator (Fur), and to be important in systemic infection of uropathogenic *E. coli* in a mouse model of infection [21–23]. YddB is distantly related to FusA in the CLANS plot, belonging to a small cluster which is in turn closely associated with a larger cluster containing sequences from pathogenic *Bordetella* and *Achromobacter* species (Fig 1, S1 Table). Interestingly, members of this *Achromobacter* cluster do not contain a FusC homologue but are commonly associated with a cytochrome *c* peroxidase protein.

The structure of YddB was solved to a resolution of 2.4 Å by X-ray crystallography by molecular replacement (S2 Table; Fig 3A). Based on identification of structural homologues using the Dali server [24], YddB is most similar to FusA with a RMSD of 2.2 Å and a Z-score

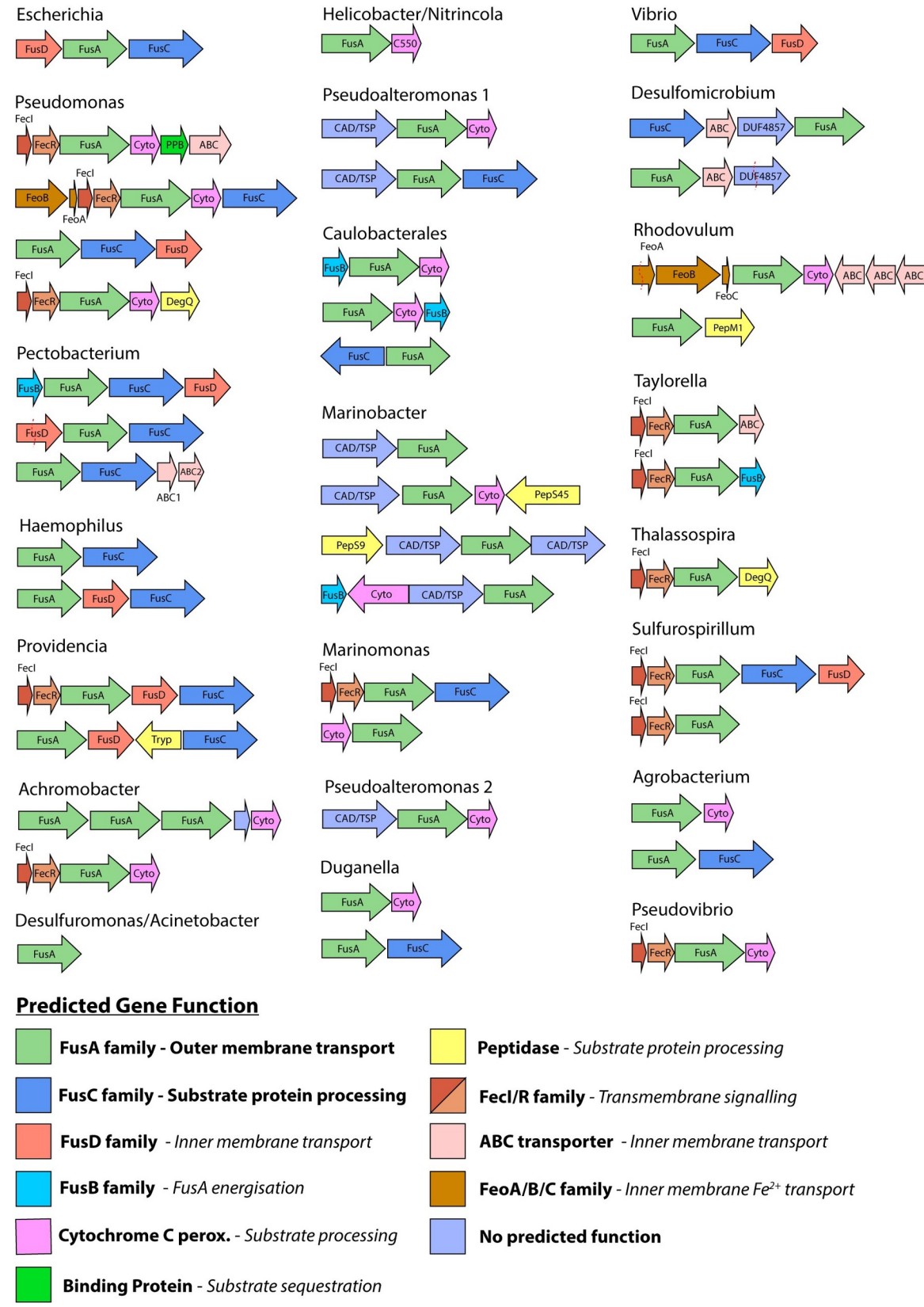

**Predicted Gene Function**

- **FusA family - Outer membrane transport**
- **FusC family - Substrate protein processing**
- **FusD family** - *Inner membrane transport*
- **FusB family** - *FusA energisation*
- **Cytochrome C perox.** - *Substrate processing*
- **Binding Protein** - *Substrate sequestration*
- **Peptidase** - *Substrate protein processing*
- **FecI/R family** - *Transmembrane signalling*
- **ABC transporter** - *Inner membrane transport*
- **FeoA/B/C family** - *Inner membrane Fe$^{2+}$ transport*
- **No predicted function**

**Fig 2. Representative genomic context of FusA homologues.** The genetic organisation of FusA homologues identified in HMMER search is shown. Labelled arrows represent genes that were present adjacent to FusA in the sequence clusters outlined in Fig 1. Genes homologous to Fus operon members are labelled accordingly, while other genes are labelled according to the function of their closest characterised homologue. A predicted function for genes is provided below, predictions in bold are based on the function of proteins from the Fus, while predictions in italics are based on function of homologous proteins. A full list of genetic context for FusA homologues, where available, is listed in S1 Table.

of 42.2 (S3 Table). As with FusA, YddB possesses a 22-stranded β-barrel fold, the pore of which is occluded by a globular N-terminal plug-domain. This fold is characteristic of the integral outer membrane TBDT family and like these proteins YddB possesses a hydrophobic transmembrane region (S2 Fig). Comparison of the structure of YddB and FusA by dissection of their extracellular loops confirmed that the two proteins are structurally analogous (Fig 3B); the eleven loops of these proteins share a common structure that is distinct from those of two other TBDTs, FhuE and Fiu (S3 Fig). Substrate capture by TBDTs is mediated with an external binding site composed of these extracellular loops [25]. The capture of the large ferredoxin substrate by FusA was shown to be mediated by a large open extracellular binding site [7]. Analysis of the YddB structure revealed that it possesses an analogous binding site, with dimensions (27.8 × 29.2 Å) capable of accommodating a bulky protein substrate of similar dimensions to ferredoxin (Fig 3C and 3D).

## PqqL is a periplasmically localised metalloprotease expressed under iron limiting conditions

We hypothesized that the uncharacterised protein PqqL, encoded in an operon with the *yddB* gene, is a protease that processes the substrate for this system upon import into the bacterial periplasm. We tested two hypotheses: (i) given the substrate of the *ydd/pqqL* operon is likely to be an iron containing protein, PqqL production will be increased in response to iron limitation; (ii) in order for PqqL to function as a processing protease, it will be periplasmically localised. The first of these hypotheses is supported by the fact that the *ydd/pqqL* operon has been shown to be regulated by Fur and is upregulated under iron limiting conditions [22, 23].

To explore the regulation of PqqL expression, we raised an antibody to recognize PqqL and performed cell fractionation and immunoblotting on the model *E. coli* strain BW25113 and the uropathogenic strain *E. coli* CFT073. As predicted, low levels of PqqL were detected in cells grown under iron replete conditions in LB medium (Fig 4A). However, the addition of the iron chelator 2'2-bipyridine (BP) to the medium led to increased expression of PqqL in both BW25113 and CFT073 (Fig 4A). No band corresponding to PqqL was detected in a Δ*pqqL* mutant *E. coli* BW25113, under any condition, confirming the specificity of PqqL detection (S4 Fig). Expression of PqqL was then tested during growth in human urine, with both BW25113 and CFT073 cells exhibiting an elevated expression relative to growth in LB media (Fig 4A). Expression of PqqL in CFT073 cells grown in urine was generally higher than BW25113, however this was variable between experiments (Fig 4A), possibly attributable to variation in sample composition. As urine is also iron limiting [26], the increase in PqqL expression is consistent to the Fur regulation of the *ydd/pqqL* operon and the observation that in CFT073, the operon is transcriptionally upregulated during growth in urine and is important for virulence [21].

To determine the cellular location of PqqL, we fractionated *E. coli* BW25113 cells grown under iron limiting conditions (100 μM BP). Immunoblot analysis of these fractions demonstrated that PqqL is localised to the periplasmic fraction, showing the same fractionation profile as the known periplasmic chaperone SurA (Fig 4B) [27]. PqqL was not detected associated with the outer membrane, fractionating distinctly from the known outer membrane protein

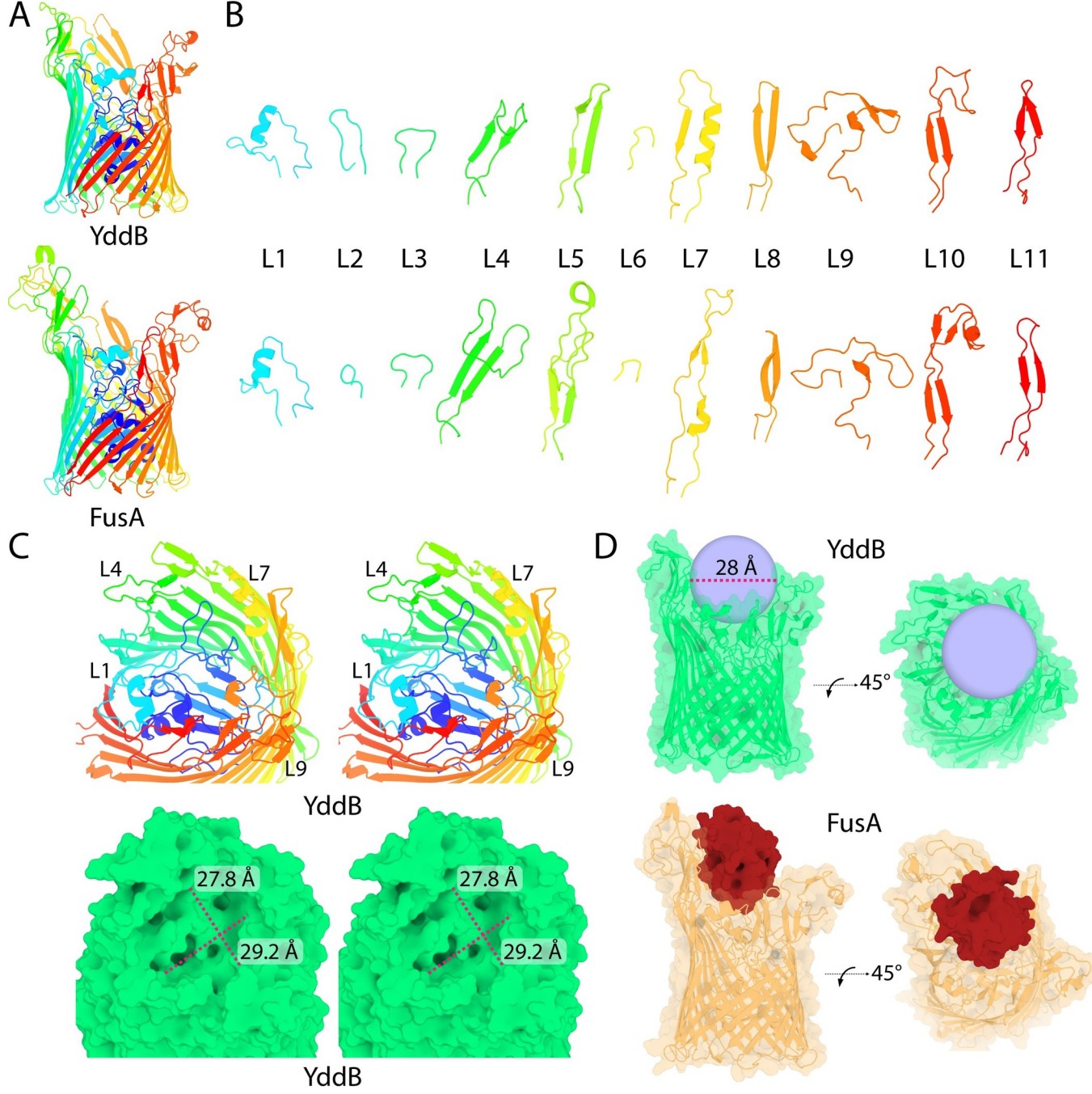

**Fig 3. The crystal structure of YddB reveals structural homology to FusA, implying conserved function in protein import.** (A) A cartoon representation of YddB (top) and FusA (bottom) showing structural homology between the two proteins despite limited (24%) sequence identity. (B) Dissection of the eleven extracellular loops of YddB (top) and FusA (bottom), showing that they conform to similar length and structural patterns. (C) A stereo view of the extracellular binding pocket of YddB (top = cartoon, bottom = green surface) showing that it consists of a large cavity capable of binding a small globular protein, in common with FusA. (D) Surface/cartoon representation of YddB (top), showing its extracellular binding pocket can accommodate a globular molecule ~28 Å in diameter. The structure of FusA docked with its ferredoxin substrate (bottom) is shown for reference.

BamA [28]. The cytoplasmic protein YtfP was absent from the periplasmic or outer membrane fractions, demonstrating the lack of cytoplasmic contaminants in these fractions (Fig 4B) [29].

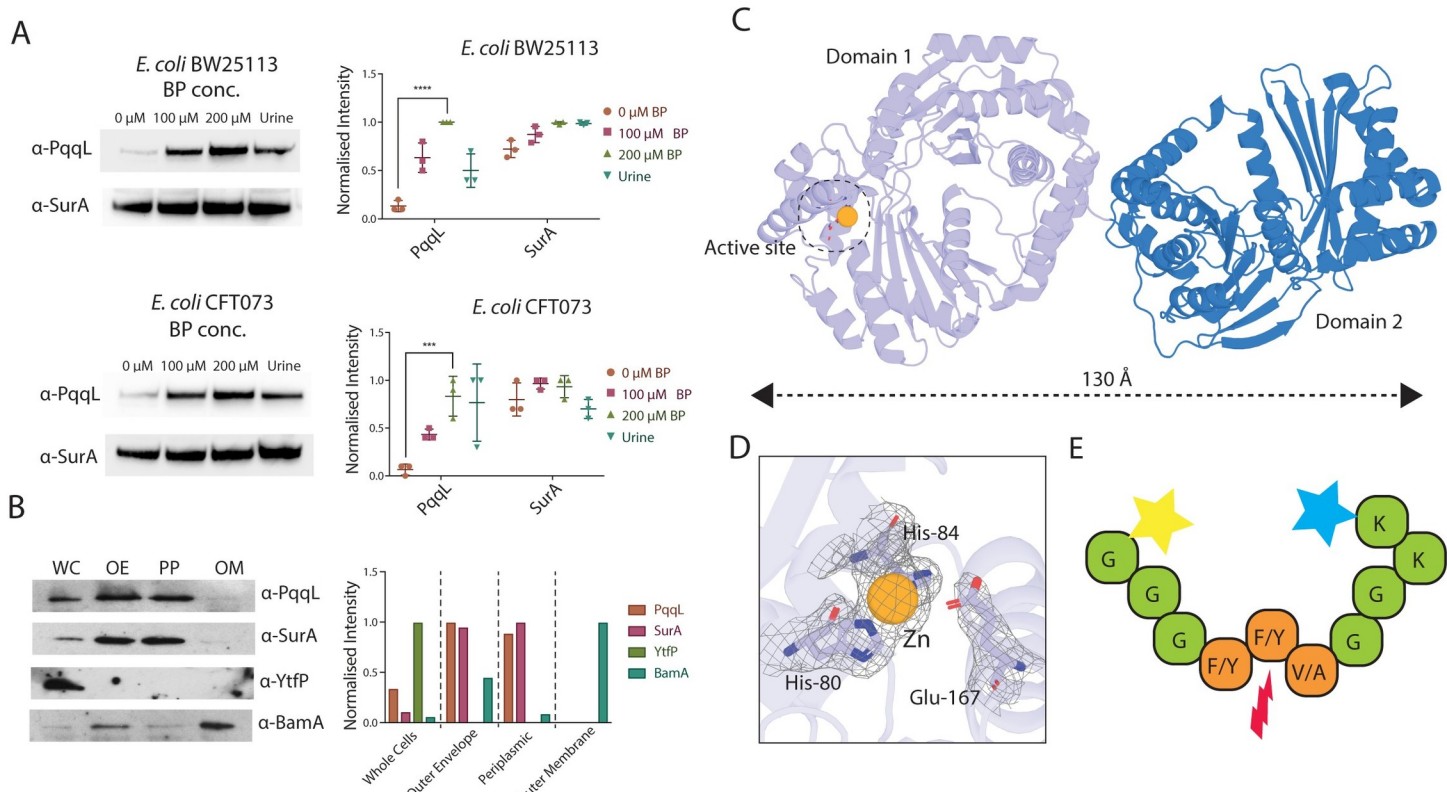

**Fig 4. PqqL is an iron regulated, periplasmically localized, metallopeptidase, with an elongated conformation.** A) An anti-PqqL western blot of *E. coli* BW25113 or CFT073 whole cells grown in the presence or absence of the iron chelator 2'2-bipyridine (BP) or in human urine. Detection of SurA is shown as a loading control. Left shows a representative blot, right shows normalised intensity of blots from 3 biological replicates. This shows more PqqL is produced under iron-limiting conditions. Indicated significant differences between conditions are based on student's t-test (* p ≤ 0.05; ** p ≤ 0.01; *** p ≤ 0.001; **** p ≤ 0.0001). B) A western blot of cell fractions from *E. coli* BW25113 grown under iron-limiting conditions (100 μM BP), showing the distribution of PqqL in whole cell (WC), outer envelope (OE) and periplasmic (PP) fractions, but not associated with the outer membrane (OM). Controls using antisera recognizing SurA (periplasmic), BamA (outer membrane) or YtfP (cytoplasmic) are shown. Left shows a representative blot, right shows quantified intensity of this blot C) The crystal structure of PqqL showing that its two 'clam shell' domains adopt a highly elongated conformation in the crystal structure, engaging in minimal intra-molecular contacts. D) The crystal structure of PqqL illustrates the presence of a putative Zn ion in the protease active site of the enzyme. E) Peptidase screening assays shows that PqqL is an active peptidase with specificity for peptides containing a F/Y, F/Y, V/A motif (Full data shown in S4 Table).

Consistent with a periplasmic localisation, immunoprecipition of PqqL from the cell lysate, followed by N-terminal sequencing revealed cleavage of the predicted signal peptide (S5 Fig).

PqqL was expressed recombinantly, purified and crystallised in order to solve its structure (S2 Table). PqqL consists of two clam-shell like halves, each formed from two M16 protease subunits (Fig 4C). This domain structure is analogous to its distant homologue FusC; however, FusC was crystallised in complex with its ferredoxin substrate and adopted a closed clam-shell conformation [6]. In contrast, in the absence of substrate, PqqL adopts an entirely open, highly extended conformation (Fig 4C). This domain arrangement has not previously been observed in structures of M16 proteases [30–32]. In addition to electron density for the PqqL polypeptide chain, density was observed in the predicted peptidase active site of the N-terminal subunit of PqqL, which was attributable to a catalytic metal ion coordinated by histidines 80 and 84 (Fig 4D). In common with other proteases of this family, these residues form part of the catalytic inverzincin H-x-x-E-H motif in which the two histidine residues coordinate a zinc ion, with the glutamate additionally required for catalysis [33]; this suggests the protein binds a putative Zn ion and serves as a metalloprotease.

To test for proteolytic activity and specificity, PqqL was subjected to peptide hydrolysis screening. In this assay, from a pool of 512 tripeptide combinations, PqqL cleaved peptides containing a tripeptide motif composed of F/Y, F/Y and V/A at positions 1, 2 and 3 respectively (Fig 4E, S4 Table). In control experiments, FusC was also shown to be active towards a distinct subset of peptides (S4 Table). PqqL was also tested for proteolytic activity towards a number of potential substrate proteins, including human ferredoxins and globins, as well as plant ferredoxin which is the substrate for FusC (S6 Fig). PqqL did not exhibit activity towards any of these proteins, suggesting they are not the substrates for the YddB/PqqL operon. This narrow specificity of PqqL towards peptide and protein substrates is consistent with that observed for FusC, which was shown to cleave plant ferredoxin but not a number of other small proteins [8]. These data demonstrate that PqqL is an active, periplasmically localised protease with a narrow substrate specificity, supporting the hypothesis that this protein cleaves a discrete imported protein substrate and is functionally analogous to FusC.

## PqqL and YddB do not support growth during iron limitation in LB media

The transcriptional upregulation of the *ydd/pqql* operon and increased expression of PqqL under iron limitation suggest that this operon may play a general role in iron acquisition [22]. To test this we constructed strains containing a genetic deletion of *yddB* and *pqqL* in *E. coli* BW25113, and a *yddB* deletion in a previously constructed strain lacking all TBDTs with a role in iron acquisition [25]. We tested these mutants for growth defects in both iron-replete and iron-limited LB media. All strains grew at the same rate as their corresponding wild-type background and reached a comparable final cell density, demonstrating that neither YddB or nor PqqL plays a significant role in iron acquisition under these conditions (Fig 5). This result suggests that the *ydd/pqql* operon plays a niche specific role in iron acquisition from a discrete protein substrate.

## Both PqqL and FusC undergo large scale conformational changes in solution

The elongated open conformation of PqqL resolved by X-ray crystallography is intriguing. In all structures previously solved, M16 family proteases adopt a closed or partially closed conformation, including the structure of FusC [6]. In order to determine if the crystal structure of PqqL is consistent with its conformation in solution we utilised small angle X-ray scattering (SAXS) (Fig 6A, S5 Table). In solution, PqqL was found to have maximum dimensions of 140 Å (Fig 6B and 6D), in good agreement with the 130 Å observed in the crystal structure (Fig 4C). In addition, as was previously demonstrated for FusC [6], PqqL was found to exhibit flexibility in solution (Fig 6C), with the simulated scattering of the crystal structure of PqqL representing a poor fit for the SAXS data (Fig 6E). Consistent with this flexibility, during crystallisation of PqqL, an additional poorly diffracting crystal form was obtained. In these crystals, a difference in orientation of the two domains of PqqL is observed, compared to the orientation of the refined crystal structure (Fig 6F, S7 Fig).

These data show that, while the crystal structure of PqqL is representative of its maximum dimensions in solution, the protein also possesses flexibility between its two domains. In previous work, solution scattering analysis of FusC revealed that while it adopts a closed conformation in the presence of its substrate in the crystal structure, it adopts an elongated conformation in solution with dimensions of 130 Å [6], very similar to those of PqqL observed in this work.

To obtain a clearer picture of the range of conformations PqqL can adopt, we utilised the ensemble optimization method (EOM). Applying this technique, the flexible linker between

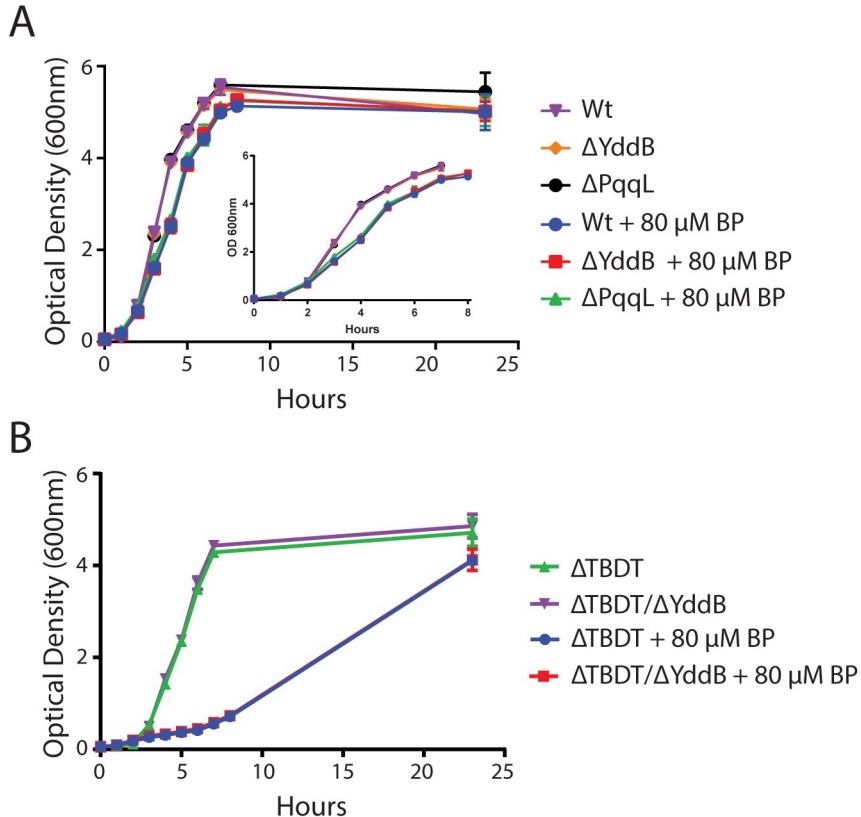

**Fig 5. Growth of Δ*yddB* and Δ*pqqL* mutant strains is identical to wildtype in iron limited LB media.** (A) *E. coli* BW25113 wildtype, Δ*yddB* and Δ*pqqL* strains were grown in LB media +/- 80 μM 2,2'bipyridine. Growth of Δ*yddB* and Δ*pqqL* strains was identical to wildtype under both conditions; all strains exhibited slower growth with 80 μM 2,2'bipyridine due to iron limitation of the media. (B) *E. coli* BW25113 ΔTBDT and ΔTBDT/Δ*yddB* were grown as strains in panel A. Growth of ΔTBDT/Δ*yddB* was identical to the ΔTBDT strain under both conditions; both strains grew slower than wildtype due to poor iron uptake ability, with very slow growth observed with 80 μM 2,2'bipyridine.

the two domains of PqqL was specified and molecular dynamics simulations were used to generate an ensemble of 10,000 models, which sampled the physically possible orientations of the two domains. A genetic algorithm was then applied to select a subset of these models that best fit the solution scattering data [34]. EOM selected an ensemble of models that produced a robust fit for the solution scattering data for PqqL (Fig 7A and 7B), providing a far better representation of observed scattering than the single model from the crystal structure (Fig 6E and 6F) [6]. The ensemble of models best representing the scattering data corresponded not only to the open conformation of the crystal structure, but also to partially closed intermediate conformations and a closed conformation analogous to that of the crystal structure of FusC in complex with ferredoxin (Fig 7A and 7C).

We therefore analysed the scattering data for a sample of FusC, and the inverse situation was observed. Along with conformations corresponding to the closed form of FusC evident from the crystal structure [6], a range of intermediate conformations were selected, along with a fully open conformation analogous to that observed in the PqqL crystal structure, and consistent with solution scattering data for FusC [6] (Fig 7B and 7D). These data provide mechanistic insight into how ferredoxin is able to enter the substrate binding cavity of FusC, as folded ferredoxin would be unable to enter the partially closed conformation observed in the crystal structure [6]. The observation that PqqL adopts an analogous range of conformational states

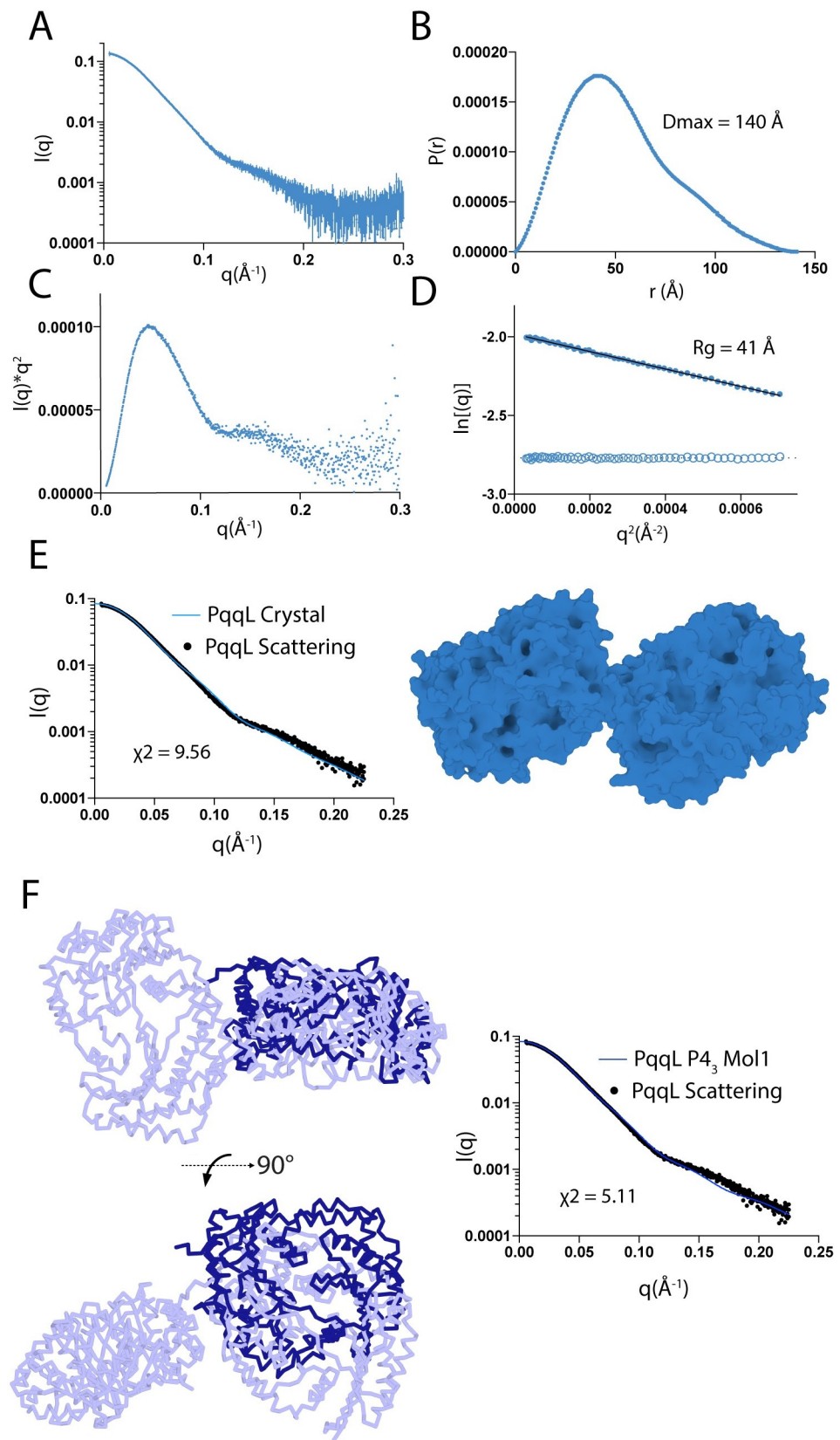

**Fig 6. PqqL is flexible and adopts an elongated conformation in solution.** Small angle X-ray scattering of PqqL (A) and derived P(r) (B), Kratky (C) and Guinier (D) plots showing that PqqL adopts a highly elongated conformation in solution with a Dmax of 140 Å and has interdomain flexibility. (E) the simulated solution scattering of the PqqL crystal structure is a relatively poor fit for solution scattering data, suggesting that PqqL adopts multiple conformations in solution. (F) PqqL adopts different conformations *in crystallo*. The light blue ribbon represents PqqL in the refined crystal structure while the dark blue ribbon represents the position of PqqL domain 2 in the low-resolution crystal form shown in S7 Fig, illustrating the difference in domain orientation between to two crystal forms. The fit of this alternative conformation is shown (right).

to FusC, which are otherwise unprecedented for proteases of this family, further supports a shared, conserved mechanism of function between PqqL and FusC.

## Discussion

The discovery of the Fus mediated ability of *Pectobacterium* species to import plant ferredoxin is a striking example of the creative strategies bacteria engage to satisfy their nutritional requirements [4, 6–8]. In this work we show that gene clusters analogous to the Fus are widely distributed in proteobacterial species. In addition, our analysis of the composition of these gene clusters reveals that they share common features, including the presence of homologues of the ferredoxin processing protease FusC and other genes predicted to be involved in

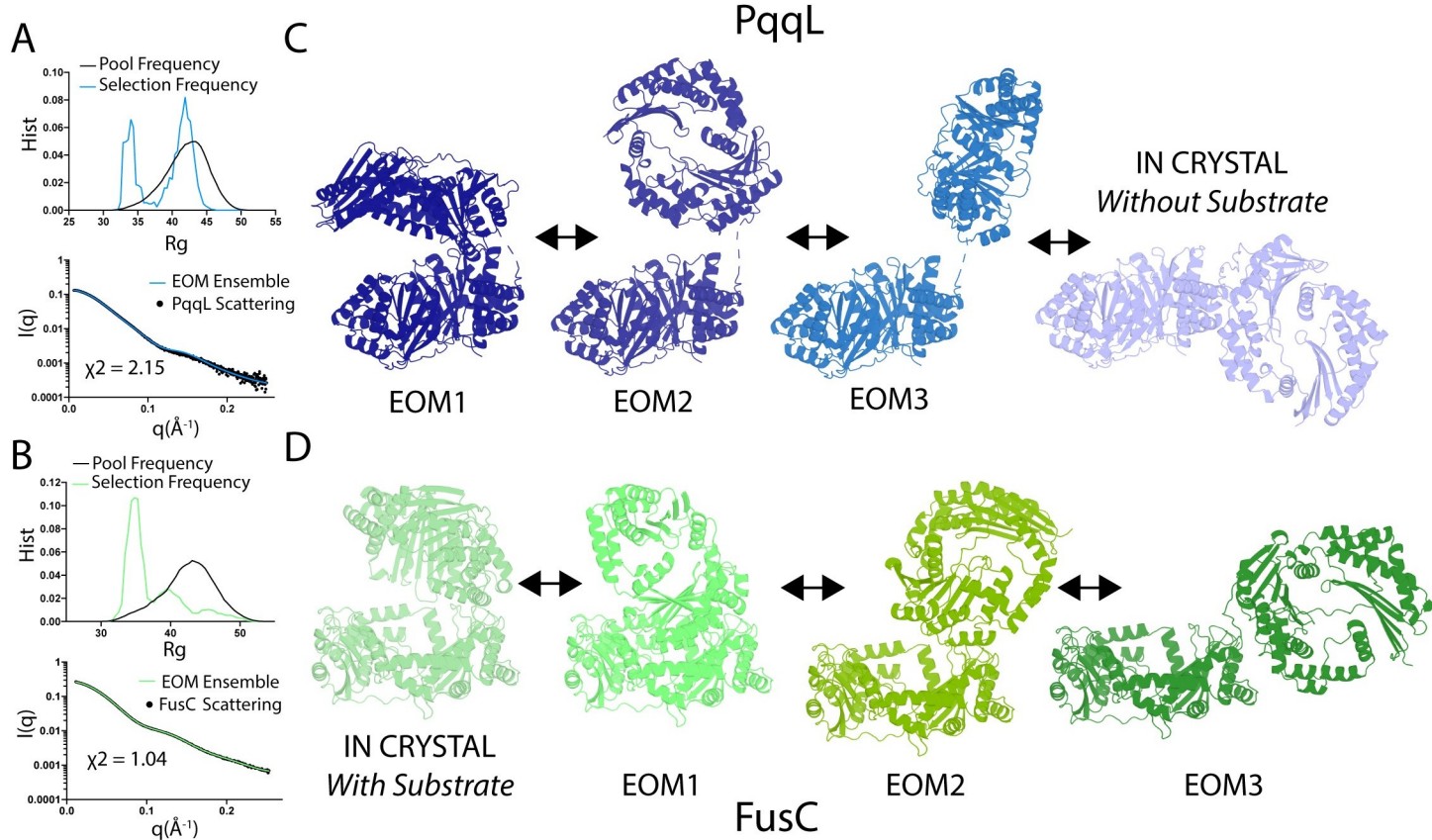

**Fig 7. PqqL and FusC sample both open and closed conformations in solution.** The radius of gyration (Rg) of the pool of models generated from EOM analysis and those selected the ensemble representative of solution scattering data for PqqL (A top) and FusC (B top). The fit of the ensemble of models selected by EOM for scattering data for experimental scattering for PqqL (A bottom) and FusC (B bottom). A selection of representative models from EOM analysis and the crystal structure of PqqL (C) and Fus (D), showing that both proteins are predicted to sample from open to closed conformations in solution.

nutrient transport. Furthermore, through structural and biochemical characterisation of the distantly related Fus homologues, YddB and PqqL from *E. coli*, we show that these proteins share key features with FusA and FusC, suggesting a conserved function. Taken together, these data provide a body of evidence that, rather than being limited to *Pectobacterium*, protein import systems are widespread in Gram-negative bacteria.

These findings add to a growing body of evidence that TDBTs are highly versatile in terms of both the size and chemical composition of the substrates they import. Initial characterisation of these transporters suggested their substrates were limited to relatively small molecules; predominantly iron binding siderophores [35]. However, their frequent targeting as receptors by antibacterial proteins hinted at the capacity for protein import [9, 36, 37]. Recent work on TDBTs distantly related to those involved in siderophore import has demonstrated their ability to transport unstructured polypeptides and polysaccharides, and even to act as exporters in a protein secretion system [38–40]. Our current work identifies a novel family of putative protein importing TBDTs for which FusA and YddB are archetypical members [7], adding further evidence to the surprising versatility the TBDT family.

In this work we show that PqqL is a protease that shares a common localisation and iron-dependent regulation with FusC [6, 8]. In addition, our structural characterisation provides intriguing clues as to the mechanism of this protease family. Both the crystal structure of FusC and the closed conformation of PqqL identified by our EOM analysis contain an internal cavity capable of accommodating a small globular protein like ferredoxin. However, in this closed conformation, the width of the opening to this cavity would prevent protein entry. By sampling between open and closed conformations in solution, these proteases could capture incoming protein substrates in an open conformation, before adopting a closed conformation for proteolysis. In addition, these conformational changes may account for the apparent ability of FusC to unfold its ferredoxin substrate, and to cleave it at multiple locations consisting of divergent amino acid sequences [6, 8]. While further characterisation is required, the striking similarities between PqqL and FusC demonstrated in this work provide the basis for the definition of a new sub-family of M16 protease.

The distribution of Fus related gene clusters in bacteria adopting such a wide variety of lifestyles, including in both marine and terrestrial environments, raises the question of what protein substrates are targeted by these systems. Given the highly divergent nature of the family, it seems unlikely that ferredoxins are the universal substrate. However, small iron-containing proteins constitute diverse and abundant protein families produced by virtually all organisms [41, 42], so the evolution of Fus-like systems targeting such proteins available in a given niche seems intuitive. For a generalist gut bacterium like *E. coli*, identification of the substrate for YddB/PqqL may prove difficult, as it could target a substrate produced by its host or a member of the complex gastrointestinal community. However, a number of species shown to possess Fus gene clusters in this work are obligate human pathogens or commensals. In these cases, it is likely that the substrate for these systems will be a host protein. By defining the structure and distribution of the Fus gene cluster family, this study paves the way for future work identifying the substrates of these systems and determining the role they play in bacterial nutrient acquisition and virulence.

## Materials and methods

### Ethics statement

All urine samples used in this study were collected with the full consent of the providing party and within the ethical guidelines of the institution where they were collected.

## Reagents and antisera generation

Human hemoglobin, cytoglobin and myoglobin and horse cytochrome C were obtained from a commercial source (Sigma-Aldrich). Polyclonal rabbit antisera for detection of PqqL, SurA, BamA and YtfP were generated at the Monash Animal Research Platform, from proteins purified in-house. Rabbits were serially injected with purified proteins in combination with complete (first injection) or incomplete (subsequent injections) Freund's adjuvant, over a period of 1–3 months, with rabbit serum periodically tested for reactivity to the target protein. Once acceptable levels of reactivity were achieved rabbits were sacrificed and serum was collected and stored in aliquots at -80˚C.

## FusA homologue identification and analysis

In order to identify FusA homologues in available bacterial genomes, a HMMER search was performed against the UniProtKB Database using FusA as the search sequence [12, 43]. No E-value cut-off was applied to hits. This search yielded 1063 sequences, which were allocated to groups with >95% sequence identity using CD-HIT [44]. One representative sequence from each group was utilised for further analysis, giving a total of 508 FusA homologue sequences. These sequences were classified by an all-against-all BLAST clustering algorithm, based on pairwise similarities. The resulting data set was visualized with CLANS with an E-value cut off of $1 \times 10^{-110}$. Sequence clusters were identified in CLANS using a network-based algorithm, with a minimum group size of 4 [13]. The sequences from each group were ordered via a sequence identity matrix. Species of origin, host of isolation and FusA genetic context were determined from genome metadata where available from the Ensemble and Uniprot databases [45, 46].

## Protein expression and purification

The open reading frames encoding YddB and PqqL were amplified by PCR from *E. coli* BW25113 using primers containing 5' NcoI and 3' XhoI restriction sites (S6 Table). They were cloned into a modified pET20b vector with a PelB signal sequence, followed by an N-terminal $10 \times$ his tag and TEV cleavage site, via restriction digestion and ligation. The resulting vector was transformed into *E. coli* BL21 (DE3) C41 cells (S6 Table) [47]. Protein expression was performed in terrific broth (12 g tryptone, 24 g yeast extract, 61.3 g $K_2HPO_4$, 11.55 g $KH_2PO_4$, 10 g glycerol) with 100 mg.ml$^{-1}$ ampicillin for selection. Cells were grown at 37˚C until $OD_{600}$ of 1.0, induced with 0.3 mM IPTG, and grown for a further 14 hours at 25˚C. Cells were harvested by centrifugation and lysed using a cell disruptor (Emulseflex) in Ni-binding buffer (50 mM Tris, 500 mM NaCl, 20 mM imidazole [pH 7.9]) for PqqL or Lysis Buffer (50 mM Tris, 200 mM NaCl [pH 7.9]) for YddB, in the presence of 0.1 mg.ml$^{-1}$ Lysozyme, 0.05 mg.ml$^{-1}$ DNAse1 and complete protease cocktail inhibitor tablets (Roche).

For PqqL, the resulting lysate was clarified by centrifugation at 30,000 g for 20 minutes and applied to Ni-agarose resin, followed by washing with $10 \times$ column volumes of Ni-binding buffer, and elution of bound proteins with a step gradient of Ni-gradient buffer (50 mM Tris, 500 mM NaCl, 500 mM Imidazole [pH7.9]) of 5, 10, 25 and 50%. Eluted fractions containing recombinant protein were pooled based on their absorbance at 280 nm and incubated with 2 mg.ml$^{-1}$ TEV protease overnight at 4˚C to cleave this his-tag. Protein was then applied to a 26/600 S200 Superdex size exclusion column equilibrated in SEC buffer (50 mM Tris, 200 mM NaCl [pH 7.9]). Eluted fractions containing PqqL, assessed by absorbance at 280 nm, were then pooled, concentrated to 10 mg/ml$^{-1}$, snap frozen and stored at -80˚C. Protein size and purity waswere confirmed by SDS-PAGE.

For YddB, the resulting lysate was clarified by centrifugation at 10,000 g for 10 min and the supernatant was then centrifuged for 1 h at 160,000 g to isolate a membrane fraction. The supernatant was decanted, and the membrane pellet was suspended in lysis buffer using a tight-fitting homogeniser. The resuspended membranes were solubilised by the addition of 10% Elugent (Santa Cruz Biotechnology) and incubated with gentle stirring at room temperature for 20 min. The solubilised membrane protein fraction was clarified by centrifugation at 20,000 g for 10 min. The supernatant containing the solubilized proteins was applied to Ni-agarose resin equilibrated in Ni-binding buffer DDM (50 mM Tris, 500 mM NaCl, 20 mM Imidazole, 0.03% Dodecylmaltoside (DDM) [pH7.9]). The resin was washed with 10–20 column volumes of Ni binding buffer DDM before elution of the protein with a step gradient of, 10, 25 and 50, 100% Ni gradient buffer DDM (50 mM Tris, 500 mM NaCl, 1 M Imidazole, 0.03% DDM [pH7.9]). YddB eluted at the 50 and 100% gradient steps, as assessed by absorbance at 280 nm. Eluted fractions containing YddB were pooled and applied to a 26/600 S200 Superdex size exclusion column equilibrated in SEC buffer DDM (50 mM Tris, 200 mM NaCl, 0.03% DDM [pH 7.9]). To exchange YddB into the detergent Octyl β-D-glucopyranoside (βOG) for crystallographic and biochemical analysis, SEC fractions containing YddB determined by absorbance at 280 nm were pooled and applied to Ni-agarose resin, equilibrated in βOG buffer (50 mM Tris, 200 mM NaCl, 0.8% βOG [pH 7.9]). The resin was washed with 10 column volumes of βOG buffer before elution with βOG buffer + 250 mM imidazole. Fractions containing YddB were pooled, and 6 × histidine tagged TEV protease (final concentration 2 mg.ml$^{-1}$) and DTT (final concentration 1 mM) were added. This solution was then dialysed against of βOG buffer at 4–6 h at 20˚C to allow TEV cleavage of the 10 × histidine tag from YddB and removal of excess imidazole. The sample was then applied to Ni-agarose resin, to remove TEV protease and the cleaved histidine-containing peptide. The flow through containing YddB from this step was collected, concentrated to 10 mg.ml$^{-1}$ in a 30 kDa cut-off centrifugal concentrator, and snap frozen and stored at -80˚C. Protein size and purify were assessed by SDS-PAGE.

The open reading frames (ORFs) for human ferredoxin 1 and 2 (minus the stop codon) were synthesised and cloned into pET21a and expressed in *E. coli* C41 (DE3). Cells were grown at 37˚C and protein expression was induced by the addition of 0.3 mM isopropyl-β-D-thiogalactoside (IPTG) at an OD600 of ∼0.6. Cultures were grown for a further 6 h at 28 ˚C. Cells were collected and resuspended in 50 mM Tris-HCl, pH 7.9, 500 mM NaCl, 20 mM imidazole, 5% glycerol, 100 μg lysozyme, and complete EDTA-free protease inhibitor cocktail tablets were added. After disruption by sonication, the supernatant was clarified by centrifugation and applied to a HisTrap-nickel agarose column equilibrated in a buffer containing 50 mM Tris-HCl, pH 7.9, 500 mM NaCl, 20 mM imidazole and 5% glycerol. Bound protein was eluted with a linear gradient of 20–250 mM imidazole in lysis buffer. Ferredoxin containing fractions were identified based on colour and analysis by SDS–PAGE, pooled and further purified using a Superdex S75 26/60 column equilibrated in 50 mM Tris, 200 mM NaCl [pH 7.9]. FusC and plant ferredoxin for peptide cleavage assays and structural analysis were purified as previously described [6, 7].

## YddB crystallisation, data collection and structure solution

Purified YddB in βOG was screened for crystallisation conditions using commercially available screens (approximately 600 conditions). Crystals grew in a number of conditions, with a condition containing 0.1 M Na cacodylate, 0.15 M Ca acetate, 15% PEG 8000 and 20% glycerol chosen for data collection. Crystals from this condition were looped and flash frozen in liquid N$_2$. Diffraction data was collected at 100 K at the Australian synchrotron and processed in the

space group $P4_12_12$ to 2.4 Å. Initial phases were obtained by molecular replacement, using phaser from the Phenix package, with a search model derived from the crystal structure of FusA from *Pectobacterium atrosepticum* SCRI1043 [4ZGV]. The crystal structure of YddB was then build and refined using Coot, Phenix package and Buster [48–50].

## PqqL crystallisation, data collection and structure solution

Purified PqqL was screened for crystallisation conditions using commercially available screens (approximately 800 conditions). Crystals grew in a number of conditions, with a condition containing 0.1 M Bis-tris propane, 0.2 M NaK tartrate, 20% PEG 3350 [pH 8.5] chosen for optimisation. Crystals initially grew with a diamond morphology and diffracted poorly to >3.2 Å in the space group $P4_3$. An additive screen was performed and it was determined that the addition of $MgCl_2$ led to a change in morphology from diamond to rectangular and an improvement in diffraction (S4 Fig). These improved crystals were cryoprotected using para-tone oil (Parabar 10312) and flash cooled in liquid $N_2$. Diffraction data was collected at 100 K at the Australian synchrotron and processed in the space group $P4_32_12$ to 2.6 Å. A partial molecular replacement solution was obtained using Phaser with an ensemble of models of the catalytic portion (approximately residues 1–230) of M16 protease structures identified by a BLAST search of the PDB; however, phases were too poor to allow further model building. Experimental phasing was attempted using selenomethionine labelled protein and numerous heavy atom-soaked crystals, but this proved unsuccessful. As an alternative, *in situ* proteolysis was undertaken by adding a 1:100 molar ratio of trypsin to PqqL prior to crystals screening. Crystals grew from this screen in 0.1 M phosphate-citrate buffer, 0.2 M NaCl, 20% PEG 8000 [pH 4.2]. Crystals were cryoprotected in the crystallisation solution plus 20% glycerol and flash cooled in liquid $N_2$. Data was collected as for full length PqqL crystals and processed in the space group $P2_1$ to 2.0 Å. A partial molecular replacement solution was obtained as with the original PqqL crystals, with two copies of the first half (AA 27 to 494) of PqqL present in the crystallographic asymmetric unit (ASU). Using these data, a model of $PqqL_{27-494}$ was built and refined using Coot, programs from the Phenix package and Buster [48–50]. $PqqL_{27-494}$ was then used as a molecular replacement model for the full length PqqL dataset, and a model of full length PqqL was built and refined using the Phenix package, Buster and Coot [48–50].

## Determination of the substrate specificity of PqqL using the Rapid Endopeptidase Profiling Library (REPLi)

The REPLi library is a combinatorial peptide library that contains 512 pools of peptides with each pool containing up to eight different variable tripeptides with the template layout of MeOC-GGXXXGG-dipicolinic acid-KK, where each X represents a variable alternative amino acid based on similar physiochemical properties, i.e. A/V, F/Y, I/L, D/E, R/K, D/E, S/T, Q/N, and P [51]. There are no Gly, His, Trp, Cys, or Met residues in the variable tripeptide region. The resulting combinations of variable tripeptides give rise to 3375 different peptides in the library in total. Methoxycarbonyl (MeOC) is the fluorophore, and dipicolinic acid is the fluorophore quencher. The soluble peptide library pools, synthesised by Mimotopes (Melbourne, Australia), contained in 512 wells in six 96-well plates were diluted using FAB to a final concentration of 50 μM. A final concentration of 1 μM PqqL was incubated with the substrate pools in FAB at 37°C. Cleavage of the substrates was monitored by measuring the increase in fluorescence intensity from the MeOC fluorophores using 55 second cycles for 30 cycles, with an excitation wavelength of 320 nm and an emission wavelength of 420 nm, using a BMG microplate reader. The initial velocity of the cleavage was indicated by the slope per unit time of the linear region of the curves.

Based on the REPLi results, 8 individual peptides from the substrate pools containing tri-peptidyl sequences of Phe/Tyr- Phe/Tyr—Ala/Val, which displayed the highest rate of cleavage by PqqL, and Ala/Val—Ala/Val—Lys/Arg displayed the highest cleavage rate for FusC, were synthesized (Mimotopes, Melbourne, Australia). To determine the values of the steady-state reaction constants, 950 nM PqqL was mixed with substrate at a range of concentrations from 0 to 600 μM and the initial velocity of reaction was plotted against the substrate concentration, allowing the determination of the $K_m$, $V_{max}$, and $k_{cat}$ values. For FusC, 1 μM was mixed with substrate at a range of concentrations from 0 to 300 μM.

## Putative substrate protein cleavage assays

Purified PqqL (1 μM) and potential substrate proteins (10 μM) were incubated in 50 mM Tris-HCl and 50 mM NaCl, pH 7.5, at room temperature. Samples were incubated for 120 min, and the reaction was stopped by the addition of SDS loading buffer. Samples were heated to 95°C for 2 min and then analyzed by SDS-PAGE.

## PqqL small angle X-ray scattering and modelling

Size Exclusion Chromatography-Small Angle X-ray Scattering (SEC-SAXS) was performed using Coflow apparatus at the Australian Synchrotron [52, 53]. Purified PqqL was analysed at a pre-injection concentration of 100 μM. Chromatography for SEC-SAXS was performed at 22°C, with an 5/150 Superdex S200 Increase column, at a flow rate of 0.4 ml/min in: 50 mM Tris, 100 mM NaCl, 5% glycerol and 0.2% sodium azide [pH7.9]. The inclusion of glycerol and azide was essential to prevent capillary fouling due to photo-oxidation of buffer components. Scattering data were collected for 1 second exposures over a $q$ range of 0.01 to 0.51 Å$^{-1}$. A buffer blank for each SEC-SAXS run was prepared by averaging 10–20 frames pre or post protein elution. Scattering curves from peaks corresponding to PqqL were then buffer subtracted, scaled across the elution peak, and compared for inter-particle effects. Identical curves (5–10) from each elution were then averaged for analysis. Data were analysed using the ATSAS package, Scatter and SOMO solution modeler [54].

## *E. coli* Δ*pqqL* and Δ*yddB* mutant generation and growth analysis

*E. coli* BW25113 Δ*pqqL* and Δ*yddB* mutant strains were created using the λ Red system [55]. Kanamycin-resistance cassettes flanked by 300 bp of genomic DNA either side of the chromosomal location of *yddB* and *pqqL* were amplified by PCR using specific mutants from the *E. coli* Keio collection [56] as templates, generating the *yddB-Kan* and *pqqL-Kan* KO cassettes. Oligonucleotide primer sequences are summarized in S6 Table. Wildtype *E. coli* BW25113 was transformed with the λ Red recombinase plasmid pKD46 [55] and grown at 30°C (LB broth + 100 μg.ml$^{-1}$ ampicillin) to an OD600 nm of 0.1 before λ Red recombinase was induced by the addition of 0.2% L-arabinose. The cultures were then grown at 30°C until an OD600 nm of 0.6–0.8 was attained and were transformed with the *yddB-Kan* or *pqqL-Kan* KO cassettes using the room-temperature electroporation method [57]. Briefly, bacterial cells were isolated by centrifugation at 3000 g for 3 min and washed twice with a volume of sterile 10% glycerol equal to the volume of culture used. The cells were then resuspended in 10% glycerol to a volume of 1/15 of that of the culture. The *yddB-Kan* or *pqqL-Kan* KO cassette DNA (100–500 ng) was then added to 100 μl of the resuspended bacteria and the mixture was electroporated. 1 ml of LB broth was added to the cells post-electroporation, and the culture was recovered at 37°C for 1 h before plating onto LB agar + 30 μg ml$^{-1}$ kanamycin. PCR was used to validate that colonies did indeed have the KanR cassette in place of the gene of interest.

To remove the KanR gene and generate "clean" *yddB* or *pqqL* deletions, the mutant strains were transformed with the plasmid pCP20 [58] containing the 'flippase cassette'. Cells were grown under either ampicillin (100 μg ml$^{-1}$) or chloramphenicol (30 μg ml$^{-1}$) selection to maintain the plasmid. A single colony of the mutant pCP20-containing strain was used to inoculate 1 ml LB broth (no selection). The culture was grown overnight at 43°C to activate expression of the flippase gene. This culture was then subjected to tenfold serial dilution in sterile LB and plated onto LB agar with no selection. The resulting colonies were patched onto LB agar containing kanamycin, chloramphenicol or no selection. PCR was used to validate colonies that, while unable to grow in the presence of kanamycin or chloramphenicol, grew in the absence of selection and had no remnant of the KanR cassette in the deletion of *yddB* or *pqqL*. The *E. coli* ΔTBDT/Δ*yddB* strain was created as above, using the previously generated multiple TBDT mutant strain *E. coli* ΔTBDT, as a starting strain [25]. *E. coli* ΔTBDT is deficient in iron uptake and grows poorly on LB agar, and this was propagated on LB agar + 100 uM Fe(II)SO$_4$.

Above mutant strains and wildtype *E. coli* BW25113 were grown in LB broth until stationary phase. These cultures were used to inoculate 20 ml of LB media +/- 80 μM 2,2-bipyridine (BP), to an OD600nm of 0.05. Cultures were grown with shaking and rate of growth was quantified by measuring OD600nm at hourly intervals.

## Detection and localisation of PqqL in *E. coli* cell extracts via western blot

The *E. coli* model strain BW25113 and the uropathogenic strain CFT073 were grown in 10 ml of LB broth with shaking overnight (S6 Table) [56, 59]. These cultures were used to inoculate 10 ml of LB broth supplemented with either 0, 100, or 200 μM 2,2-bipyridine (BP) or human urine, donated provided by the study's lead author of the study. Cultures were grown till late log phase and cells were harvested by centrifugation at 3000 g for 20 min at 4°C.

For detection of PqqL in whole cell extracts, *E. coli* were resuspended in 50 mM Tris, 200 mM NaCl pH 8.0, cell density was normalised based on OD$_{600}$ measured at harvest. Cell numbers were determined by serial dilution and colony counting, and SDS-PAGE loading buffer was added to buffer containing 4.8x10$^7$ cells, samples were then heated at 95°C for 5 min.

For cell fractionation experiments, the Tris-Sucrose-EDTA method was performed [60]. All steps were carried out on ice unless otherwise stated. Cells utilised for these experiments were grown in LB + 100 μM BP. The supernatant was carefully discarded from the sedimented cells and the last few drops were removed by pipette. Cells, where gently resuspended in 1 ml (per 100 ml of bacterial culture) of TSE buffer (200 mM Tris, 500 mM sucrose, 1 mM EDTA [pH 8.0]) using a wire loop. The cells suspension was incubated on ice for 30 min, before sedimentation of cells at 16,000 g for 30 min at 4°C. The supernatant from this step represents the outer-envelope fraction, containing both periplasmic and outer membrane components. This fraction was further centrifuged at 100,000 g to sediment the outer membranes and to yield a more homogenous periplasmic fraction as the supernatant fraction. The outer membrane fraction from this step was resuspended in a minimal quantity of TSE buffer. The protein content of the whole cell, outer envelope, periplasmic and outer membrane fractions was estimated by BSA assay, and protein concentrations were normalised by dilution. Fractions were snap frozen in liquid N$_2$ and stored at -80°C.

For PqqL and control protein detection, samples were separated on a 12% SDS-PAGE gel, which was subsequently blotted to 0.2 μm pore size nitrocellulose membrane. The membranes were blocked by incubation with TBST buffer (50 mM Tris, 150 mM NaCl, 0.1% Tween 20, [pH 7.5]) plus 5% skim milk powder (TBST-B) for 1 h at room temperature. Membranes were then incubated in TBST-B, containing a 1:5,000–1:20,000 dilution of rabbit derived anti-PqqL, anti-BamA, anti-YtfP or anti-SurA polyclonal serum for 1 h at room temperature. Membranes

were then washed thoroughly with TBST, before incubation with a 1:20,000 dilution of a HRP-conjugated anti-rabbit secondary antibody for 1 h at room temperature. Membranes were with washed thoroughly with TBST, before protein bands were visualised by chemiluminescence and imaged via X-ray film or CCD-camera. Relative band intensity was quantified from scanned X-ray film or directly from CCD-camera images, by 1-D integration of band intensity using the 'Plot Bands' tool in ImageJ [61]. For CCD-images, integrated images were captured within the dynamic range of the detector and for film multiple exposure lengths were captured and those within linear intensity range, determined by plotting the intensity of multiple exposures, were utilised for quantitation. For comparison of blots from different experiments, band intensity was normalised by utilising the formula $I/I_{max}$, where I = the intensity of the band and $I_{max}$ = the intensity of the most intense band from that blot. For blots judging expression levels of PqqL, raw PqqL band intensity was adjusted for loading variability using the formula $I_{raw}/I_{SurA}$, where $I_{raw}$ = the raw intensity of the PqqL band and $I_{SurA}$ = the normalised intensity of the SurA band from the corresponding lane.

## PqqL immunoprecipitation

PqqL was isolated from whole cell lysate of *E. coli* CFT073 grown in LB + 100 μM BP until late log phase, by immunoprecipitation. Protein A agarose beads were washed with binding buffer (50 mM Na Phosphate, 200 mM NaCl, 1mM EDTA [pH 8.0]) and incubated with rabbit derived anti-PqqL serum diluted 1:1 with binding buffer, at room temperature for 1 h. Beads were then washed extensively with binding buffer to remove serum contaminants.

*E. coli* CFT073 cells were sedimented by centrifugation at 4000 g for 10 min. The cell pellet was resuspended in lysis buffer (50 mM Tris, 150 mM NaCl [pH 8.0]) and lysed by sonication. Lysate was clarified by centrifugation at 14,000 g for 10 min. Clarified lysate was incubated with anti-PqqL loaded protein A agarose for 1 h at room temperature. The beads were washed extensively with lysis buffer, resuspended in 1 × SDS-PAGE sample buffer (62.5 mM Tris-HCl, 2.5% SDS, 0.002% Bromophenol Blue, 10 mM dithiothreitol (DTT), 10% glycerol pH [6.95]) and incubated at 95˚C. The sample was then separated on a 12% SDS-PAGE gel. A band corresponding to the size of full length PqqL (~100 kDa) was excised and N-terminal sequencing was performed by Edman degradation.

## Supporting information

**S1 Fig. Clustering analysis of representative TonB-dependent transporters.** Representative/structurally characterized TonB-dependent transporters were clustered using CLANS, demonstrating that YddB and FusA form a sequence cluster that is similarly distantly related to other TonB-dependent transporters. FusA/YddB are similarly distant to the main group of transporters as the highly divergent SusC family from *Bacteroides* spp.; further illustrating a distant relationship between FusA and other transporters. Dots represent individual sequences and grey lines represent pairwise similarity relationships. An E-value cut-off of 1e-110 was used for clustering.
(TIF)

**S2 Fig. YddB possesses a conserved TBDT β-barrel fold with a hydrophobic transmembrane region.** The crystals structure of YddB shown as rainbow cartoon (N-terminus = blue, C-terminus = red) (left), and electrostatic surface (right). The electrostatic surface illustrates the presence of a hydrophobic transmembrane region, which embeds YddB in the membrane. Octyl β-D-glucopyranoisde detergent molecules observed shielding the hydrophobic region in

the crystal structure are shown as spheres.
(TIF)

**S3 Fig. The extracellular loops of YddB are structurally distinct from TonB-dependent transporters of divergent function.** The extracellular loops of YddB (A) are distinct in structure and length from those of FhuE (B) and Fiu (C), transporters for coprogen and catecholate siderophores respectively.
(TIF)

**S4 Fig. Anti-PqqL antisera do not detect PqqL in *E. coli* BW25113 *ΔpqqL*.** (A) A representative western blot of *E. coli* BW25113 *ΔpqqL* whole cells with anti-PqqL (top) and anti-SurA (bottom) in the presence and absence of 2,2'bipyridine, showing no band corresponding to PqqL is detected in this strain. Detection of PqqL in wildtype *E. coli* BW25113 is shown as a reference. (B) Quantitation of 3 biological replicate of blots of panel A.
(TIF)

**S5 Fig. N-terminal sequencing of immunoprecipitated PqqL reveals cleavage of predicted signal peptide in vivo.** PqqL immunoprecipitated using anti-PqqL serum was isolated via SDS page (left) and N-terminally sequenced using Edman degradation. The sequence of the corresponding band (AALPQD) is consistent with the N-terminal sequence of PqqL after cleavage of its predicted signal peptide.
(TIF)

**S6 Fig. Purified PqqL does not cleave plant ferredoxin or a panel of mammalian iron containing proteins.** Coomassie brilliant blue stained SDS-PAGE gel visualisation of protease cleavage reactions containing various small iron containing proteins in the presence and absence of PqqL. No proteolytic cleavage by PqqL was observed in these substrates.
(TIF)

**S7 Fig. PqqL exhibits conformational flexibility *in crystallo*.** (A) In the absence of $MgCl_2$ PqqL formed poorly diffracting crystals in the space group $P4_3$, the addition of $MgCl_2$ led to an increase in symmetry and change in space group to $P4_32_12$. (B) PqqL molecules in crystals of the space group $P4_3$ exhibited a difference in conformation between their two domains, indicative of inherent flexibility of PqqL.
(TIF)

**S1 Table. FusA homologues identified by HMMER search using FusA as the interrogation sequence, sorted by CLANS clustering analysis.** Sequence identity of different members of each cluster (labels for the X-axis of the matrix follow those for the Y-axis). The genetic context of FusA homologues and environment of isolation of species are included where available. A, B, C, D = corresponding Fus homologues, other proteins are labelled with name of nearest homologous protein or description of conserved domain, X = unknown function.
(XLSX)

**S2 Table. Crystallographic data collection and refinement statistics for YddB and PqqL crystal structures.**
(XLSX)

**S3 Table. Dali search results for the structure of YddB.**
(XLSX)

**S4 Table. Peptides cleaved and cleavage kinetics for PqqL and FusC in peptide hydrolysis screening assay.**
(XLSX)

**S5 Table. Data collection and processing data/statistics for SAXS scattering of PqqL.**
(XLSX)

**S6 Table. Primers, plasmids and strains used in this study.**
(XLSX)

**S7 Table. Quantitation data for western blots.**
(XLSX)

## Acknowledgments

This research was undertaken on the MX1, MX2 and SAXS/WAXS beamlines at the Australian Synchrotron, part of ANSTO (CAP12312, and M12480). We would like to thank the Monash Crystallisation Facility for their assistance with sample characterisation, crystallographic screening and optimisation.

## Author Contributions

**Conceptualization:** Rhys Grinter, Trevor Lithgow.

**Data curation:** Rhys Grinter, Lakshmi C. Wijeyewickrema.

**Formal analysis:** Rhys Grinter, Pok Man Leung, Lakshmi C. Wijeyewickrema, Chris Greening.

**Funding acquisition:** Rhys Grinter, Trevor Lithgow.

**Investigation:** Rhys Grinter, Lakshmi C. Wijeyewickrema.

**Methodology:** Rhys Grinter, Lakshmi C. Wijeyewickrema, Dene Littler, Simone Beckham, Robert N. Pike, Daniel Walker, Chris Greening, Trevor Lithgow.

**Project administration:** Rhys Grinter, Trevor Lithgow.

**Resources:** Rhys Grinter, Dene Littler, Simone Beckham, Robert N. Pike, Chris Greening.

**Supervision:** Trevor Lithgow.

**Validation:** Rhys Grinter.

**Visualization:** Rhys Grinter.

**Writing – original draft:** Rhys Grinter.

**Writing – review & editing:** Rhys Grinter, Pok Man Leung, Lakshmi C. Wijeyewickrema, Dene Littler, Simone Beckham, Daniel Walker, Chris Greening, Trevor Lithgow.

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
