## [Decision Letter · Decision Letter 0]

8 Jul 2019

Dear Dr Grinter,

Thank you very much for submitting your Research Article entitled 'Protease-associated import systems are widespread in Gram-negative bacteria' to PLOS Genetics. Your manuscript was fully evaluated at the editorial level and by independent peer reviewers. The reviewers appreciated the attention to an important problem, but raised some substantial concerns about the current manuscript. Based on the reviews, we will not be able to accept this version of the manuscript, but we would be willing to review again a much-revised version. We cannot, of course, promise publication at that time.

As you will see from their comments, the three reviewers agreed that this is an interesting study that is clearly written and that it presents new insight into this protease-associated import system. However the three reviewers agreed also that the manuscript is missing a bit more experimental and biological results to support the structural work and its resultant conclusions. Each of the reviewers pointed to missing experiments. For instance the authors should confirm that that the Ab does not bind to similar region of FusC, they should determine if (separate, non-polar) mutations in YddB and PqqL impair the ability of E. coli to grow in low-iron, as this would be indicative of a role in iron import, cellular fractionation, immunoblotting should be done for YddB as it was for PqqL (Fig 4B) and/or microscopy for surface localization. Furthermore the Western blots should be quantified and crystallographic statistics after anisotropy correction for YddB, especially cc1/2 and I/s and the validation reports for all the structures, need to be provided.

If you decide to revise the manuscript for further consideration at PLOS Genetics, please aim to resubmit within the next 60 days, unless it will take extra time to address the concerns of the reviewers, in which case we would appreciate an expected resubmission date by email to plosgenetics@plos.org.

[LINK]

We are sorry that we cannot be more positive about your manuscript at this stage. Please do not hesitate to contact us if you have any concerns or questions.

Yours sincerely,

Carmen Buchrieser

Associate Editor

PLOS Genetics

Josep Casadesús

Section Editor: Prokaryotic Genetics

PLOS Genetics

Reviewer's Responses to Questions

**Comments to the Authors:**

Reviewer #1: Grinder and colleagues are investigating the structure and function of the uncharacterised proteins YddB and PqqL from Ecoli that show some similarity to the FusC and FusA proteins that are involved in iron acquisition from ferredoxin. FusA and FusC encode for an outer membrane receptor that uptakes ferredoxin whereas FusC is a peptidase that releases iron from ferredoxin. Using sequence analysis they show that YddB and PqqL show similarity to the Fus system. Cell growth assays under limiting iron conditions shows that the proteins are upregulated. They also solved the crystal structure of both proteins and they show very similar fold to the Fus proteins suggesting that they are likely distant homologues. Functional data, revealed that YddB has protease activity.

Overall, they provide good evidence that these systems are well conserved among proteobacteria.

Some issues that need to be addressed:

1. The authors show that under limit iron conditions, the PqqL and YddB are over expressed. Considering the high degree of similarity between PqqL and FusC, can they be confident that the Ab does not bind to similar region of FusC. A control experiment whereas a knockout of PqqL in the presence of Abs should be performed to strengthen this claim. They should also provide information on how the Abs were raised.

2. The authors have nicely shown that PqqL displays protease activity using a peptide screen. Since they claim that this system is homologous to FusC, why not perform the activity in the presence of ferredoxin? Do the identified peptide sequences much the ferredoxin sequence that they could map them on?

3. I am concerned with the high Rmerge values for all data sets at low resolution. Is it possible that the data suffer from pseudo symmetry? Is the redundancy for PqqL full length really 125?

The authors should provide crystallographic statistics after anisotropy correction for YddB, especially cc1/2 and I/s.

In the revision they should provide the validation reports for all the structures.

They should also list Ramachandran statistics.

Reviewer #2: Grinter et al. recently discovered and characterized a unique class of protein import systems dedicated iron uptake from ferredoxin in Pectinobacterium. The system consists of an outer membrane TonB-dependent porin FusA and a periplasmic protease FusC. Here, by using a combination of bioinformatic, biochemical and structural analyses, these authors show the presence of functionally analogous and structurally similar systems in a range of proteobacteria. To support their bioinformatics analysis, they determined the structure of a related system from Escherichia coli comprising the outer membrane component YddB and a periplasmic protease PqqL. They show that PqqL is induced upon iron limitation in E. coli, supporting the role of the system in iron scavenging from an iron-containing protein. PqqL structure determination and its comparison to that of FusC reveals a protein composed of two domains connected by a short linker. These domains adopt a closed conformation in the presence of substrate and an extended one in its absence. This conformational transition is thoroughly characterized by SAXS and molecular dynamics analysis in PqqL and FusC. The authors also determine the substrate specificity of PqqL and demonstrate that it is rather narrow, in line with its role in cleavage of a specific substrate.

The study is original, well executed and the article is very clearly written. The study reveals important information on this new class of systems, by showing that many proteobacteria have the capacity to take up proteins from the environment. The study therefore provides a basis for a vast field of research that might reveal other biological functions of these protein import systems.

Minor comments :

1. The PqqL western blots appear to be nonlinear and there is a clear difference in protein and control levels between the two strains in Fig. 4A. The recent guidelines require that the linear range of detection be determined and the Western blots be quantified to support the claim that there is more PqqL in urine than in the presence of BiP. The loaded samples correspond to how many bacteria?

2. I suggest that the ion in Fig. 4C be depicted with a different color for better contrast. In addition, the color of the zoomed area in Fig. 4D should be same as in Fig. 4C.

3. Could the authors describe what was their positive control in the FRET assay for peptide specificity?

4. Lines 133-139. The authors advance a claim that bacteria containing these protein import systems tend to associate with plant or animal hosts, a claim immediately contradicted by their presence in marine bacteria. It may be better to avoid any general claims at this point as too little is known about their functions or “specific lifestyles”. As for most TonB-dependent transporters, these systems are likely to promote uptake of scarce nutrients from the environment.

Other minor text comments:

line 71: … a bacterium …

l. 163: …distinct from …

l. 476: Cells were…

SI legends: l. 727: similarly distant from…

l. 734: Structurally distinct…

l. 736: remove “the siderophores” at the end of the line.

l. 741: … N-terminally sequenced…

l. 755: sequence identity of different members…

Reviewer #3: Previous work done with Pectobacterium spp. had shown that the outer membrane protein FusA and the periplasmic protein FusC proteins conjoin to import and then degrade ferredoxin as a means toward iron assimilation. In this very interesting follow-up, Grinter et al show that i) gene clusters (proteins) related to FusA/FusC exist in many types of Proteobacteria, ii) the structure of the E. coli protein YddB is similar to that of FusA, and iii) E. coli PqqL is a periplasmic protease that is induced by low-iron growth conditions and is structurally similar to FusC. The MS is very well written and interesting. The structural biology work that was done is especially impressive. The MS’ conclusions are generally appropriate. Thus, the findings here have implications for many Gram-negative’s, including both plant and animal pathogens. However, the MS would benefit from the inclusion of more “biology” (points 1 & 2) and genetic analysis (point 3), in order to strengthen the conclusions made.

Major points

1. Given that the homologs of YddB and PqqL are involved in iron assimilation and that the current study finds PqqL to be more highly expressed in low-iron, there should be some attempt to determine whether YddB-PqqL promotes iron assimilation in E. coli. (That past work by others had shown that YddB is important in systemic infection by a strain of UPEC does not alone make this point.) It is true, as the authors mention in their Discussion, that the substrate for the system need not be the same as that of the Pectobacterium system (i.e., ferredoxin); however, at the least, the authors should determine if (separate, non-polar) mutations in YddB and PqqL impair the ability of E. coli to grow in low-iron, as this would be indicative of a role in iron import. The fact that PqqL is hyper-expressed in LB containing the iron chelator BP (Fig 4A) strongly suggests that the proteins are needed under these growth conditions. It might be necessary to mutate yddB and pqqL in a strain that is lacking siderophore in order to clearly / dramatically see a role for YddB and PqqL.

These experiments are worthwhile even if they do not reveal a link to growth in low-iron, as this would provide evidence for an import that is rather distinct from the Pectobacterium system.

2. Lines 147-156. Although it was shown later in the MS that PqqL is localized to the periplasm compatible with its role as an analog of FusC, the outer membrane / surface localization of YddB was not documented. But, it should have been, given the (implied) conclusion that YddB is an outer membrane transporter analogous to FusA. Cellular fractionation and immunoblotting could be done as it was for PqqL (Fig 4B) and/or microscopy for surface localization.

3. Lines 171, 182-184. Given that levels of PqqL are increased in low-iron growth conditions, it should be determined and discussed whether the yddB/pqqL operon is iron-regulated and Fur-regulated. Basic qRT-PCR can determine if the genes are iron-regulated, and sequence analysis should be able to identify a putative Fur box. (The fact that the operon was shown by others to be upregulated in urine does not alone make this point.). Following on point 2, immunoblotting could then confirm whether YddB levels are also influenced by iron levels.

Minor points

1. Lines 220-221, 260. Materials & Methods needs a section on how FusC was obtained and used.

2. Line 338. YddB is missing from the section header.

3. Lines 341 464. Insert references to Table S6.

4. Lines 354, 370, 376. Make clearer how the fractions of interest were identified.

5. Line 426. Provide a reference and source for this reagent and method.

6. Lines 466-467. Provide a source for the human urine used.

7. Lines 491-492. Provide the source and if needed methods used to obtain these five antisera.

8. Lines 522-651. The formatting of the references is not consistent.

9. Lines 689, 692. Should this read “BW251113” rather than “K12”?

10. Lines 138, 727, 735, 755. Check for typos and word usage.

**Have all data underlying the figures and results presented in the manuscript been provided?**

Reviewer #1: Yes

Reviewer #2: Yes

Reviewer #3: Yes

PLOS authors have the option to publish the peer review history of their article (what does this mean?). If published, this will include your full peer review and any attached files.

Reviewer #1: Yes: Konstantinos Beis

Reviewer #2: No

Reviewer #3: No

---

## [Decision Letter · Decision Letter 1]

19 Sep 2019

Dear Dr Grinter,

We are pleased to inform you that your manuscript entitled "Protease-associated import systems are widespread in Gram-negative bacteria" has been editorially accepted for publication in PLOS Genetics. Congratulations!

Yours sincerely,

Carmen Buchrieser

Associate Editor

PLOS Genetics

Josep Casadesús

Section Editor: Prokaryotic Genetics

PLOS Genetics

Comments from the reviewers (if applicable):

Reviewer's Responses to Questions

**Comments to the Authors:**

Reviewer #1: The authors have addressed all my comments. The additional data have improved the manuscript.

minor comment:

Ensure that the YddB resolution is reported as 2.4A throughout the manuscript and not as 2.0A.

Since the YddB crystallographic data are not anisotropically corrected, the authors should remove this statement in Table S2: 'b Correction applied using 'Diffraction Anisotropy Server'

Similarly, in materials and methods section 'YddB Crystallisation, Data Collection and Structure Solution' remove this statement: 'Crystals diffracted anisotropically and so data was elliptically

truncated using the anisotropy server to a* = 2.5, b* = 2.5 and c* = 2.0 Å'

Reviewer #2: The authors have addressed my comments thouroughly. I have no further issues with the revised version.

Reviewer #3: The authors have adequately addressed all of my previous concerns by performing additional experiments, and updating the text with more explanations, detail, and/or references. I do not have any further concerns or comments. This is an improved and now solid MS.

**Have all data underlying the figures and results presented in the manuscript been provided?**

Reviewer #1: Yes

Reviewer #2: Yes

Reviewer #3: Yes

PLOS authors have the option to publish the peer review history of their article (what does this mean?). If published, this will include your full peer review and any attached files.

Reviewer #1: Yes: Konstantinos Beis

Reviewer #2: No

Reviewer #3: No

**Data Deposition**

http://datadryad.org/submit?journalID=pgenetics&manu=PGENETICS-D-19-00927R1

**Press Queries**

---

## [Editor Report · Acceptance letter]

26 Sep 2019

PGENETICS-D-19-00927R1 

Protease-associated import systems are widespread in Gram-negative bacteria 

Dear Dr Grinter, 

We are pleased to inform you that your manuscript entitled "Protease-associated import systems are widespread in Gram-negative bacteria" has been formally accepted for publication in PLOS Genetics! Your manuscript is now with our production department and you will be notified of the publication date in due course.

With kind regards,

Matt Lyles

PLOS Genetics

On behalf of:
